

# Graph coloring using the reduced quantum genetic algorithm

Sebastian Mihai Ardelean and Mihai Udrescu

Computer and Information Technology, University Politehnica of Timisoara, Timisoara, Timis, Romania

## ABSTRACT

Genetic algorithms (GA) are computational methods for solving optimization problems inspired by natural selection. Because we can simulate the quantum circuits that implement GA in different highly configurable noise models and even run GA on actual quantum computers, we can analyze this class of heuristic methods in the quantum context for NP-hard problems. This paper proposes an instantiation of the Reduced Quantum Genetic Algorithm (RQGA) that solves the NP-hard graph coloring problem in $O(N^{1/2})$. The proposed implementation solves both vertex and edge coloring and can also determine the chromatic number (*i.e.*, the minimum number of colors required to color the graph). We examine the results, analyze the algorithm convergence, and measure the algorithm's performance using the Qiskit simulation environment. Our Reduced Quantum Genetic Algorithm (RQGA) circuit implementation and the graph coloring results show that quantum heuristics can tackle complex computational problems more efficiently than their conventional counterparts.

## INTRODUCTION

In general, the heuristic methods—and genetic algorithms (GA) in particular—have received a great deal of research interest, being one of the most straightforward and widely applied forms of evolutionary computation *Spector (2004)*. Indeed, GAs represent one of the most known methods for solving optimization problems. In the context of classical computation and conventional dedicated hardware, GAs achieve only marginal performance improvements over the deterministic perspectives. Therefore, quantum computing appears as one of the possible improvement solutions to approach NP-complete problems (*Udrescu, Prodan & Vlăduțiu, 2006*).

Simulated genetic operators (such as mutation and crossover) and population dynamical processes (such as reproduction and selection) underpin the GA. In the simplest form, the candidate solutions to search or optimization problems are encoded in arrays, referred to as chromosomes. The algorithm begins with a randomly generated initial population of chromosomes, and then it evolves the population over multiple generations in search of an optimal solution. A genetic algorithm has four phases: initialization, selection, reproduction, and termination. Concisely, a GA is a search for the optimal solution represented by the individual chromosome with the highest fitness value.

Corresponding author
Sebastian Mihai Ardelean,
sebastian.ardelean@student.upt.ro

On the other hand, as stated in *Nielsen & Chuang (2002)* and in *Spector (2004)*, quantum computation is very powerful for solving various problems due to its specific properties and phenomena such as entanglement, interference, and exponential parallelism. As such, the merge between genetic algorithms and quantum computing is natural and beneficial (*Udrescu, Prodan & Vlăduțiu, 2006*).

This paper aims to provide a method of instantiating the Reduced Quantum Genetic Algorithm (RQGA) to solve the graph coloring problem. Our implementation solves the most recognizable form of graph coloring, namely, vertex coloring (*Kwok & Pudenz, 2020*). The problem requires coloring each vertex of a graph such that no two adjacent vertices have the same color. Furthermore, our approach can also solve the edge coloring problem and find the chromatic number of a graph.

Accordingly, our main contributions to the state-of-the-art are defined by the following facts:

I. RQGA is merely a framework and not a specific algorithm. Thus, our first contribution is to instantiate the framework for the graph coloring problem and provide design solutions:

    a) We provide an original design of the fitness function such that it accepts valid and invalid individuals as arguments. (An invalid individual has a chromosome configuration that does not satisfy given conditions).

    b) We design the Oracle quantum circuit such that the search for the highest fitness will not take place in the invalid individuals area.

    c) We provide a solution for individual and fitness representation, inspired by the one proposed by *Fabrikant & Hogg (2002)*.

II. Our RQGA-based solution is a purely quantum method:

    a) Our implementation solves the graph coloring problem in $\mathcal{O}(\sqrt{N})$ oracle queries.

    b) The same implementation solves both the node and edge coloring forms of the graph coloring problem.

    c) Our implementation determines the chromatic number of a graph.

## BACKGROUND

### Quantum computing

Classical computers are built according to the laws of classical physics. Therefore, a full specification of its state can be specified by a measurable set of numbers. According to *Häner et al. (2016)*, sustaining the pace of Moore's law has become increasingly difficult such that an alternative should be considered to meet the power and performance requirements; one of the most prominent such alternatives is quantum computing.

According to *Spector (2004)*, quantum computing describes computational processes that rely for their efficacy on information processing hardware having quantum mechanical properties. Thus, as mentioned in *Nielsen & Chuang (2002)*, quantum computers offer an essential speed advantage over their classical equivalents.

While in classical computers the bit is the basic information unit, the qubit is the analogous concept for quantum computers; as mentioned in *Udrescu-Milosav (2009)*, qubits are the fundamental information storage unit. Like the classical bit that has a state (either 0 or 1), the qubit can be in either $|0\rangle$ or $|1\rangle$, which correspond to the classical ones, respectively. Additionally, the qubit can also be in a superposition state

$$|\psi\rangle = \alpha|0\rangle + \beta|1\rangle, \tag{1}$$

where

$$\alpha, \beta \in \mathbb{C}, |\alpha|^2 + |\beta|^2 = 1, \tag{2}$$

until it is observed/measured. As presented in *Nielsen & Chuang (2002)*, the first postulate of quantum mechanics states that the state of a quantum system is described by a unit vector in a complex Hilbert space, $\mathscr{H}$. Thus, a system of $n$-qubits (referred as quantum register) has $2^n$ computational basis states of the form $|x_0 x_1 \ldots x_{n-1}\rangle$, and its quantum state is a normalized vector in $\mathscr{H}^{2^n}$,

$$|\psi\rangle = \sum_{i=0}^{2^{n-1}} \alpha_i |i\rangle, \sum_{i=0}^{2^{n-1}} |\alpha_i|^2 = 1. \tag{3}$$

According to the second postulate of quantum mechanics, the evolution of a closed quantum system is described by a unitary transformation. Considering that $|\psi_1\rangle$ is the quantum state of the system at time $t_1$, $|\psi_1\rangle$ is related to the quantum state $|\psi_2\rangle$ at time $t_2$, $t_1 < t_2$, by a unitary operator $U$. The unitary transformation $U$ applied by a quantum computer with $n$-qubits, also called a gate, is represented by a unitary matrix in $\mathbb{C}^{2^n \times 2^n}$.

As presented in *Nannicini (2020)*, unitary matrices are norm-presenting; considering a unitary matrix $U$ and a vector $x$, $\|Ux\| = \|x\|$. Therefore, for a $n$-qubit system, the quantum state $|\psi_1\rangle \in \mathbb{C}^{2^n}$ is an unitary vector and the result of applying $U$ onto state $|\psi_1\rangle$,

$$|\psi_2\rangle \rightarrow U|\psi_1\rangle, U \in \mathbb{C}^{2^n \times 2^n}, \tag{4}$$

is the unitary vector $|\psi_2\rangle \in \mathbb{C}^{2^n}$, leading to the observation that quantum operations are linear and reversible.

According to *Barenco et al. (1995)*, any unitary transformation can be expressed as a composition of gates. Thus, infinitely many quantum gates can be constructed. A finite set of quantum gates are universal for quantum computation since any unitary transformation may be approximated by a quantum circuit involving only those gates. In Table 1 we present some basic quantum unitary transformations we use in this paper, along with the unitary matrix and the graphical representation used in the implementation of the RQGA.

The third postulate of quantum mechanics states that quantum measurement is described by a measuring transformation of the quantum state (*i.e.*, the measurement operator $M_m$). For a quantum state $|\psi\rangle$, the probability that the outcome of the measurement is $m$ is $P(m) = \langle \psi M_m^\dagger M_m|\psi\rangle$.

The single qubit $|\psi\rangle = \alpha|0\rangle + \beta|1\rangle$ where $\alpha, \beta \in \mathbb{C}$ and $|\alpha|^2 + |\beta|^2 = 1$ is projected by the measurement onto the basis $\{|0\rangle, |1\rangle\}$, thus yielding the outcome $|0\rangle$ with probability $\alpha^2$ or

**Table 1 Quantum unitary transformations (*i.e.*, gates) used in the implementation of the RQGA.**

| Gate | Graphical representation | Unitary matrix | Description |
|---|---|---|---|
| Hadamard | $\boxed{H}$ | $H = \dfrac{1}{\sqrt{2}} \begin{bmatrix} 1 & 1 \\ 1 & -1 \end{bmatrix}$ | $\lvert 0 \rangle \rightarrow \dfrac{1}{\sqrt{2}}(\lvert 0 \rangle + \lvert 1 \rangle)$ and $\lvert 1 \rangle \rightarrow \dfrac{1}{\sqrt{2}}(\lvert 0 \rangle - \lvert 1 \rangle)$ |
| Pauli-X | $\boxed{X}$ | $X = \begin{bmatrix} 0 & 1 \\ 1 & 0 \end{bmatrix}$ | $\lvert 0 \rangle \rightarrow \lvert 1 \rangle$ and $\lvert 1 \rangle \rightarrow \lvert 0 \rangle$ |
| Pauli-Y | $\boxed{Y}$ | $Y = \begin{bmatrix} 0 & -i \\ i & 0 \end{bmatrix}$ | $\lvert 0 \rangle \rightarrow i\lvert 1 \rangle$ and $\lvert 1 \rangle \rightarrow -i\lvert 0 \rangle$ |
| Pauli-Z | $\boxed{Z}$ | $Z = \begin{bmatrix} 1 & 0 \\ 0 & -1 \end{bmatrix}$ | $\lvert 1 \rangle \rightarrow \lvert -1 \rangle$ and $\lvert 0 \rangle$ is left unchanged |
| Controlled-Not | $c \bullet c$ <br> $t \oplus t$ | $CNOT = \begin{bmatrix} 1 & 0 & 0 & 0 \\ 0 & 1 & 0 & 0 \\ 0 & 0 & 0 & 1 \\ 0 & 0 & 1 & 0 \end{bmatrix}$ | $\lvert c \rangle \lvert t \rangle \rightarrow \lvert c \rangle \lvert t \otimes c \rangle$, where $\lvert c \rangle$ is the control qubit and $\lvert t \rangle$ is the target *Nielsen & Chuang (2002)* |

$\lvert 1 \rangle$ with probability $\beta^2$. By measurement, the quantum state is irrevocably disturbed and collapsed to the specific Hilbert space basis state, consistent with the measurement result. Therefore, for any basis states $\lvert a \rangle$ and $\lvert b \rangle$ and the quantum state $\lvert \psi \rangle = \alpha \lvert a \rangle + \beta \lvert b \rangle$ with $\lvert \alpha \rvert^2 + \lvert \beta \rvert^2 = 1$ expressed in terms of the orthonormal base $\{\lvert a \rangle, \lvert b \rangle\}$, the measurement can be performed with respect to the $\lvert a \rangle$ and $\lvert b \rangle$ basis and the outcome will be $\lvert a \rangle$ with probability $\alpha^2$ or $\lvert b \rangle$ with probability $\beta^2$.

According to *Nielsen & Chuang (2002)*, two principles are applied to measurement: the **Principle of deferred measurement** and the **Principle of implicit measurement**. The principle of deferred measurement states that the measurement can be moved from an intermediate stage of the circuit to its end. The principle of implicit measurement states that any unterminated quantum wires may be assumed as measured at the end of the circuit.

As presented in *Nielsen & Chuang (2002)*, the fourth postulate describes the state space of a composite physical system as the tensor product of the state spaces of the component physical system.

## Genetic algorithms and quantum computing

The literature proposes several quantum genetic algorithms, from algorithms that combine operations running on classic computers with quantum operators to genuine quantum evolutionary algorithms (*Lahoz-Beltra, 2016*). Using evolutionary algorithms for synthesizing quantum circuits has been thoroughly investigated and relevant progress has also been reported in the field of Quantum-Inspired Genetic Algorithm (QIGA) (*Ruican et al., 2007*).

In *Ruican et al. (2007)*, the authors propose a method of synthesizing quantum circuits using genetic programming. The chosen approach is to split the potential circuits into vertical and horizontal levels used for chromosome definition. Furthermore, *Ruican et al. (2008)* propose an object-oriented framework for genetic algorithms in quantum circuit synthesis.

In *Gepp & Stocks (2009)*, the authors analyze evolutionary algorithms for synthesizing quantum algorithms. They claim that genetic algorithms and genetic programming have been used with great success to analyse and optimize quantum algorithms. Moreover, by successfully evolving small new quantum algorithms, scalable quantum algorithms are proven.

In *Malossini, Blanzieri & Calarco (2008)*, the authors propose a quantum genetic algorithm called Quantum Genetic Optimization Algorithm (QGOA). The algorithm starts with a qubit representation of the population and a quantum evaluation unit. Then, the selection step of the GA uses the quantum selection procedure, while the remaining steps (*e.g.*, crossover, mutation, and substitution) are performed on a classical computer.

*Nowotniak (2010)* briefly presents Quantum-Inspired Evolutionary Algorithm (QIEA). Moreover, in *Nowotniak & Kucharski (2010)* the authors present the results after applying a meta-optimization algorithm for tuning the QIEA parameters for numerical optimizations problems coded in real numbers. The algorithm used for meta-optimization is based on local unimodal sampling and is applied to adjust the crossover rate and the contraction factor. According to *Nowotniak & Kucharski (2010)*, the results show that the local unimodal sampling is an effective method for the meta-optimization of QIEAs.

*Zhang (2011)* presents a survey of the research in QIEAs; the author shows the differences between different solutions and compares the advantages and limitations of the various solutions. The author introduces the *Binary Observation* QIEA, shows that the use of Q-gates as a variation operator instead of crossover, recombination and mutation have a positive impact on the optimality of the solutions. Compared to *Binary Observation* QIEA, *Binary Observation* QIEA with crossover and mutation employs the mentioned operators to replace the migration operator with benefits in population diversity, especially in the later stages of evolution. *Binary Observation* QIEA with the novel update method for Q-gates, defines the Q-gate angle $\theta = kf(\alpha,\beta)$, which directly influences the convergence speed. According to *Zhang (2011)*, the algorithm also introduces the catastrophe operator; the scope of the new operator is to replace the best individual with the best individual of a new population if the best solution remains unchanged over some generations.

*SaiToh, Rahimi & Nakahara (2014)* introduces a novel quantum genetic algorithm with quantum crossover operation applied to all chromosomes in parallel. The proposed solution uses two identical copies of a superposition corresponding to a generation to relabel the qubits. They also show that the quantum genetic algorithm with quantum crossover operations achieves a quadratic speedup over its classical counterpart.

*Kumar & Kumar (2018)* propose a novel Quantum-Inspired Evolutionary View Selection Algorithm (QIEVSA). The authors also bring forward an experimental comparison with other evolutionary view selection algorithms. The method makes use of the *Binary Observation* QIEA algorithm to select the Top-K views from a multidimensional lattice. The authors experimentally show that QIEA is able to select good quality Top-K views for higher dimensional data sets (*Kumar & Kumar, 2018*).

In *Moussa, Calandra & Humble (2019)*, the authors present an iterative quantum algorithm for finding the maximum value of a function along with the corresponding implementations. The approach utilizes quantum search and extends the RQGA with a dynamic oracle function.

*Lahoz-Beltra (2016)* presented a comparison between three algorithms: Quantum Genetic Algorithm (QGA), Hybrid Genetic Algorithms (HGA), and RQGA. QGA and HGA represent classical optimization methods, inspired by the principles of quantum computing. The first QGA step is the initialization of a quantum population of chromosomes; each chromosome is defined by a string of qubits. Hadamard gates and rotation around Y-axis with a random number process each qubit. Then, a classical computer performs the fitness evaluation, while a quantum computer (employing rotation gates) performs the selection. The quantum computer also performs mutation using inversion gates. Likewise, an HGA implements the above steps, the difference being that the crossover operator is also quantum.

## Graph coloring applications

As mentioned in *Mahmoudi & Lotfi (2015)*, the graph coloring problem is a well-studied NP-hard problem because of its multiple applications that include timetabling, scheduling, radio frequency assignment, computer register allocation, printed circuit board testing, and so forth.

The authors of *Bincy & Presitha (2017)* present the application of graph coloring in scheduling problems, such as timetable scheduling, aircraft scheduling, and seating plan design. In *Dondi, Fertin & Vialette (2011)*, the authors present the use of graph coloring in pattern matching.

In *Hennessy & Patterson (2018)*, the authors also present the application of graph coloring in register allocation algorithms; to this end, they build a graph representing all possible candidates for allocation to a register.

In *Orden et al. (2018)*, the authors present the application of graph coloring to WI-FI channel assignment. Therefore, for a given spectrum of colors and a matrix of interferences between each pair of colors, the authors use the Threshold Spectrum coloring problem for fixing the number of colors available to minimize the interference threshold. Moreover, they use the Chromatic Spectrum Coloring and a given threshold to find the smallest number of colors that respect the constrain.

In *Demange et al. (2015)*, the authors present the applications of graph coloring in routing and wavelength assignment, dichotomy-based constrained encoding, frequency assignment problems, and scheduling. As the authors mentioned, the routing and wavelength assignment problem involves generating a set of light paths for each request—routing—and selecting a light path per request, and assigning wavelengths to the selected path–wavelength assignment. To solve the problem, the optical network is transformed into a graph and the wavelengths are assigned using graph coloring. The dichotomy-based constrained encoding problem can be reduced to a graph coloring problem and, as mentioned by the authors, is used to generate asynchronous implementations of finite state machines.

Therefore, considering the large number of applications, solving graph coloring has received a great deal of research interest *Dokeroglu & Sevinc (2021)*. For instance, in *Aragón Artacho & Campoy (2018)*, the authors present a method for solving the graph coloring problem using the Douglas-Rachford Algorithm. Indeed, they prove that the algorithm is an effective heuristic for solving this NP-hard problem.

In *Dokeroglu & Sevinc (2021)*, the authors propose a novel memetic Teaching-Learning-Based Optimization (TLBO) algorithm, combined with tabu search, for solving the graph coloring problem. Furthermore, they developed a version of TLBO that makes use of parallelism for solving large graphs. The authors claim that the results of the parallel version of the algorithm are better than those of its sequential counterpart, and the solution is competitive with state-of-the-art solutions presented in the literature.

*Mahmoudi & Lotfi (2015)* present a new approach for solving the graph coloring problem using a discrete Cuckoo Optimization Algorithm (COA). As stated by authors, the success rate of the proposed solution is above 60% for solving DIMACS (Center for Discrete Mathematics and Theoretical Computer Science) benchmark graphs, while in most cases is close to 100%.

In *Tomar et al. (2013)*, the authors propose a novel artificial bee colony optimization algorithm for solving the graph coloring problem and compare the solutions with other algorithms such as first fit, largest degree based ordering, and saturation degree based ordering. They claim that the proposed solution can converge in a few iterations and optimally allocate colors to the vertices of a graph. As mentioned by the authors, compared to the other algorithms, their proposed solution can converge to the optimal solution in very few iterations. Moreover, they show that the performance of the artificial bee colony optimization algorithm improved with the increase of the graph size.

## Graph coloring in quantum computing

Lately, quantum heuristic solutions for the graph coloring problem have received a great deal of interest. Most of the proposed solutions rely on quantum implementations of simulated annealing. In *Kudo (2018)*, the authors use the Constrained Quantum Annealing (CQA) to solve the graph coloring problem with the advantage of reducing the Hilbert space dimension. Also, in *Tabi et al. (2020)* the authors introduce a space-efficient embedding for quantum circuits that solve the graph coloring problem and present the performance gain for this method. The authors indicate the limitation of the existing Quantum Annealing (QA) hardware solutions by running various numerical simulations and comparing results obtained with standard and enhanced Quantum Approximate Optimization Algorithm (QAOA) circuits.

*Titiloye & Crispin (2011)* propose comparing Classical Annealing and QA in solving graph coloring problems. The QA algorithm used in the comparison utilizes the Path-Integral Monte Carlo for Quantum Annealing—a population-based extension to simulated annealing inspired by quantum mechanics *Titiloye & Crispin (2011)*. According to the authors, the QA algorithm outperforms its classical counterpart, and even finds the best algorithm solutions.

In *Silva et al. (2020)*, the authors start by formulating the Quadratic Unconstrained Binary Optimization Problem (QUBO)—a powerful mathematical tool that can map any problem to a quantum annealing computer. *Silva et al. (2020)* solve the problem using different approaches: QUBO with classical simulated annealing in a simulated quantum environment, using a D-Wave quantum machine, and reducing polynomial degree using both the D-Wave library and their implementation. The results show that both Simulated Annealing and QA produce good heuristics for the graph coloring problem, although more solutions can be found using the quantum approach.

*Kwok & Pudenz (2020)* propose a comparison between a heuristic graph coloring approximation algorithm—based on QA—and a fully classical implementation. The metrics for calculating performance are success probability, wall clock time, and time-to-solution. For wall clock time and time-to-solution, the quantum solution performs better than its classical equivalents. As mentioned in *Kwok & Pudenz (2020)*, the classical algorithm takes significantly longer to return a graph coloring for all graph sizes. The same is also true in the case of time-to-solution. The probability of success for the classical algorithm is lower than the quantum algorithm in the case of smaller size graphs. As the authors mention, the results of their experiments suggest a potential quantum advantage (*Kwok & Pudenz, 2020*).

*Fabrikant & Hogg (2002)* also present a quantum computer heuristic search algorithm for graph coloring. The authors consider the NP-complete case of 3 colors. For the problem representation, the authors introduce the idea of associating each node with a value from 0 to 3 (using two bits per node). As mentioned in *Fabrikant & Hogg (2002)*, 0 represents an uncolored node, and any other value represents a specific color assigned to the node. Since the generalized Hamming distance underpins the quantum algorithm, using this representation has the benefit that the distance between the states is a simple function of their bit strings.

*Shimizu & Mori (2021)* present an exponential-space quantum algorithm computing the chromatic number using the Quantum Random Access Memory (QRAM); the authors also describe a polynomial-space quantum algorithm not using QRAM for the 20-coloring problem. Their main result is the theorem that states that, to solve the chromatic number problem, the running time for the exponential-space bounded error quantum algorithm using QRAM is $\mathcal{O}(1.9140^n)$ (*Shimizu & Mori, 2021*).

## METHODOLOGY

Quantum computing promises substantial speedups over conventional machines in many practical applications. To this end, the Qiskit toolchain fosters the development and simulation of quantum algorithms and applications that will run on real quantum machines *Anis et al. (2021)*. As described in *Wille, Van Meter & Naveh (2019)*, Qiskit is an end-to-end open-source software library for quantum computing, covering the full stack from the actual interaction with the IBM Q hardware up to the application-level algorithms. Compared to other quantum simulators, Qiskit Aer allows the execution of algorithms on noiseless or noisy simulators, so that we observe the expected results, or the effects of noise on the expected results.

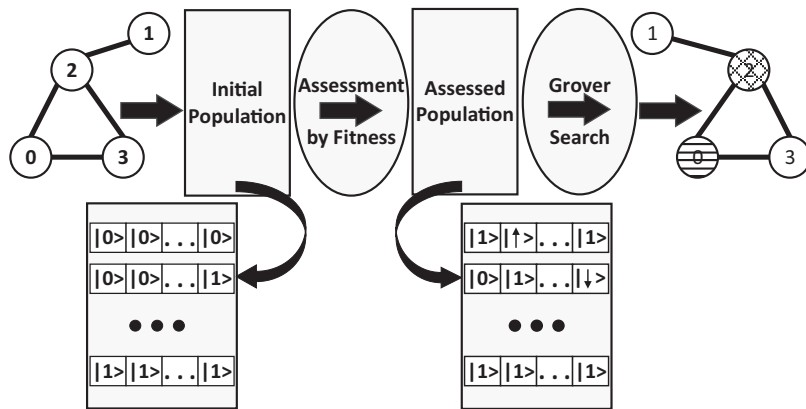

**Figure 1 The overview of applying RQGA to solve the graph coloring problem: we start with a randomly generated graph, create the superposition of $n \times |V|$ basis states (representing the initial population made of valid and invalid individuals) then apply the fitness function over the individual register.** The assessed population consists of invalid individuals represented as negative numbers in two's complement and valid individuals represented as positive numbers (also in two's complement). Grover Search is applied over the assessed population. The outcome of the algorithm are the valid configurations for graph coloring.

Therefore, we decided to instantiate the RQGA for solving the node coloring version of the graph coloring problem. On randomly generated graphs, we employ RQGA to color the nodes such that no two adjacent nodes have the same color (see Fig. 1). Our goals are to observe the results, measure the performance of the algorithm in a simulated environment, and use the results for further algorithm development.

The Reduced Quantum Genetic Algorithm is an entirely quantum evolutionary algorithm proposed by *Udrescu, Prodan & Vlăduțiu (2006)*, which puts forth a new methodology for running genetic algorithms on a quantum computer (*Lahoz-Beltra, 2016*). They also provide a design for the special-purpose oracle that works with a modified version of the maximum finding algorithm (*Ahuja & Kapoor, 1999*). Thus, the proposed method reduces the quantum genetic algorithm to a Grover search. The RQGA takes advantage of the fact that the best fitness value can be marked without destroying the superposition of the quantum register. In this context, Grover's algorithm (*Grover, 1996*) can find the solution of the problem. The main steps of RQGA are:

- Initialize a superposition of all possible chromosomes
- Compute the unitary operation corresponding to fitness computation
- Apply Grover's algorithm:

  - Ask the oracle $O$ to mark the fitness value bigger than some value *max*
  - Apply Grover's diffusion operator $G$ to augment the quantum amplitude of the marked fitness values

- Perform the measurement to get one of the marked fitness values
- Update *max* value with the measured fitness value

- Repeat the steps above until the *max* value is not improved
- Return the chromosome value corresponding to *max* as solution.

## Problem statement

We consider an undirected graph $G = (V,E)$ where $V$ is the set of nodes and $E$ is the set of edges. We define $K$ as the set of colors. Our problem is to find the best way of assigning the colors in $K$ to nodes from $V$, such that no two adjacent nodes have the same color ($e_{ij} \in E$ with $k(v_i) \neq k(v_j)$ for distinct nodes $v_i$ and $v_j$). Thus, coloring $G$ is the mapping $k : V \rightarrow E$ such that $k(v_i) \neq k(v_j)$ if $\exists e_{ij} \in E$, as mentioned in *Titiloye & Crispin (2011)*. The chromatic number of the graph, $\chi(G)$, can be found by first detecting the coloring for a high estimate of $\chi(G)$ and then successively narrowing the available colors.

## Test graphs

Let the $K$ be the set of colors and $G = (V,E)$ a randomly generated Erdős-Rényi graph with a defined edge probability (*Wang & Chen, 2003*). In $G$, the number of edges in the graph is $|E|$, and $|V|$ is the number of nodes.

## Implementation

The first step initializes all individual–fitness register pairs $|u\rangle_i \otimes |0\rangle_i$ as

$$|\psi\rangle_1 = \frac{1}{\sqrt{2^N}} \sum_{i=0}^{2^N-1} |u\rangle_i \otimes |0\rangle_i. \tag{5}$$

The representation we use for graph coloring uses (*Fabrikant & Hogg, 2002*) work; the difference is that we consider each binary combination a color. If the number of colors used for coloring the graph is not a power of 2, then the unused combinations are considered invalid. The chromosome is a $(n \times |V|)$-qubit quantum register, where each color is represented using $n$-qubits. Since there are $2^n - 1$ possible colors, we define a subset $F \subset K$ of invalid color combinations. The quantum chromosome is a superposition of all $(n \times |V|)$-bit classical chromosome values, representing valid and invalid individuals, see *Udrescu, Prodan & Vlăduțiu (2006)* (an invalid individual represents a combination that contains at least one of the invalid colors codes). The same approach can be used for edge coloring with the difference that the chromosome is a $(n \times |E|)$-qubit quantum register. Algorithm 1 shows how to create the initial population.

We represent the individual using a $(n \times |V|)$-qubit individual quantum register and the fitness value using a $M$-qubit fitness register (see Fig. 2). As such, our algorithm uses $2^{n \times |V|} - 1$ quantum register pairs. In order to maintain correlation between each individual and its corresponding fitness value, the fitness function must be an unitary operator, $U_{fit}$, corresponding to the Boolean function $f : \{0, 1\}^{n \times |V|} \rightarrow \{0, 1\}^M$.

The next step is to calculate the fitness value for all individuals. The step is achieved by defining the function $f : \{G, K\} \rightarrow \mathbb{N} \cup \{-1\}$, $G = (V, E)$ as

---

**Algorithm 1** Quantum circuit initialization.

1: Create the individual quantum register $|u\rangle$

2: Create the fitness quantum register $|fitness_u\rangle$

3: Create the carry quantum register $|carry\rangle$

4: Create the oracle quantum register $|oracle\rangle$

5: Create the positive number of edges quantum register $|val\rangle$

6: Create the quantum circuit $QC$

7: $|u\rangle = H|u\rangle$

8: $|oracle\rangle = H|oracle\rangle$

---

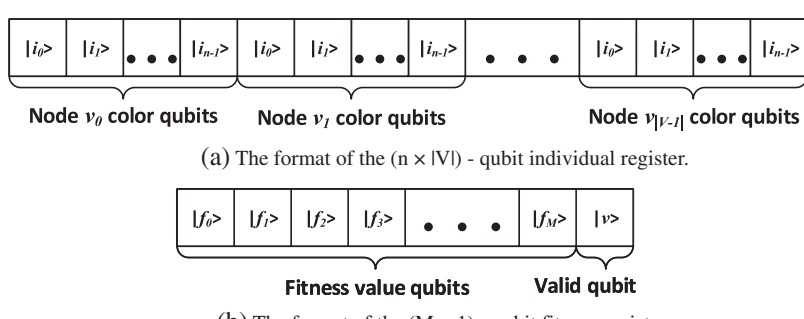

(a) The format of the (n × |V|) - qubit individual register.

(b) The format of the (M + 1) - qubit fitness register.

**Figure 2 (A) The format of the individual register.** A $n$-qubit binary combination of the color is assigned to each node, such the size of the register is $(n \times |V|)$-qubits. (B) The format of the fitness register: $M$ qubits used for fitness value representation in two's complement, and 1 qubit used to indicate the validity of the corresponding chromosome.

$$f((V,E),k) = \begin{cases} -1 & k \in F \\ 0 & \nexists k(v_i), k(v_j) \in K, k(v_i) \neq k(v_j), e_{ij} \in E \\ x & \exists k(v_i), k(v_j) \in K, k(v_i) \neq k(v_j), e_{ij} \in E, x \in \mathbb{N}. \end{cases} \tag{6}$$

$K$ is a set of colors

$$K = \{x_{n-1}\ldots x_i\ldots x_{00}, x_{n-1}\ldots x_i\ldots x_{01}, \ldots x_{n-1}\ldots x_i\ldots x_{0j}, \ldots, x_{n-1}\ldots x_i\ldots x_{0|V|-1}\}, \tag{7}$$

where $x_i \in \{0,1\}$ is a single bit and $x_{n-1}\ldots x_i\ldots x_{0j}$ is the binary representation of a color from the set $K$. The fitness function returns '−1' if the individual is invalid and '0' if there are no two adjacent nodes with different coloring. If there are adjacent nodes with different coloring, then the fitness function will return $x \in \mathbb{N}$, representing the number of edges between those nodes that met the criteria.

For solving the edge coloring problem, we can apply the same fitness function with minimal adjustments. Thus, the fitness function $f : \{G, K\} \to \mathbb{N} \cup \{-1\}$, $G = (V, E)$, returns '−1' if the individual is invalid (the chromosome contains at least one of the invalid colors), '0' if there are no two incident edges with different coloring and $x \in \mathbb{N}$, representing the number of nodes with incident edges of different coloring. Equation

$$f((V,E),k) = \begin{cases} -1 & k \in F \\ 0 & \nexists k(e_{ij}), k(e_{ik}) \in K, k(e_{ij}) \neq k(e_{ik}), e_{ij}, e_{ik} \in E \\ x & \exists k(e_{ij}), k(e_{ik}) \in K, k(e_{ij}) \neq k(e_{ik}), e_{ij}, e_{ik} \in E \end{cases} \quad (8)$$

defines the fitness function for the edge coloring problem.

In both Eqs. (6) and (8), the fitness function accepts both valid and invalid individuals as arguments. As mentioned in *Udrescu, Prodan & Vlăduțiu (2006)*, the values of the fitness function represented in the two's complement belong to distinct areas in the quantum state vector representation, corresponding to valid and invalid individuals. Negative fitness values represent invalid individuals, with the most significant bit indicating the validity ('0' on the most significant bit position represents an invalid individual and '1' represents a valid one). Thus, $U_{fit}$ characterized by the fitness function $f$ is an unitary operator

$$U_{fit} : |u\rangle \otimes |0\rangle \rightarrow |u\rangle \otimes |f_{fit}(u)\rangle, \quad (9)$$

where $|u\rangle \otimes |0\rangle$ is the individual-fitness value quantum pair register. After applying

$$U_{fit} : \frac{1}{\sqrt{2^N}} \sum_{i=0}^{2^N-1} |u\rangle_i \otimes |0\rangle_i \rightarrow \frac{1}{\sqrt{2^N}} \sum_{i=0}^{2^N-1} |u\rangle_i \otimes |f_{fit}(u)\rangle_i \quad (10)$$

on the initial population, we obtain an assessed population

$$|\psi\rangle_2 = U_{fit}|\psi\rangle_1 = \frac{1}{\sqrt{2^N}} \sum_{i=0}^{2^N-1} |u\rangle_i \otimes |f_{fit}(u)\rangle_i. \quad (11)$$

We implement the fitness sub-circuit using $n$-qubit Controlled-Not gates. In Fig. 3, we present the $U_{fit}$ sub-circuit with input and output qubits, and in Algorithm 2 we present the description of the subcircuit.

According to the algorithm presented in *Udrescu, Prodan & Vlăduțiu (2006)*, the next step involves the random generation of value $max \in \mathbb{N}, max > 0$ from the interval $[2^{M+1}, 2^{M+2} - 1)$, with $M$ representing the size of the fitness quantum register, such that the search for the individual with the highest fitness will not occur in the invalid individuals area.

In the implementation of the graph coloring problem, an individual is invalid if it contains at least one of the invalid colors; the fitness value is −1 in that case. On the other hand, the fittest individual contains the configuration that colors the biggest number of nodes from $G$ such that no two adjacent nodes $v_i, v_j$ have the same color. The fitness value of the fittest individual represents the number of edges between adjacent nodes with different coloring. Thus, instead of the randomly generated *max* value, we can use the number of edges in the graph, and so the search of the highest fitness will occur in the valid individuals area.

In the next step, we apply the Oracle and Grover diffuser $m - 1$ times, where $m$ is the number of Grover Iterations.

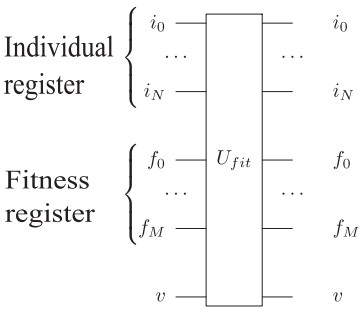

(a) Fitness assessment sub-circuit represented by $U_{\text{fit}}$.

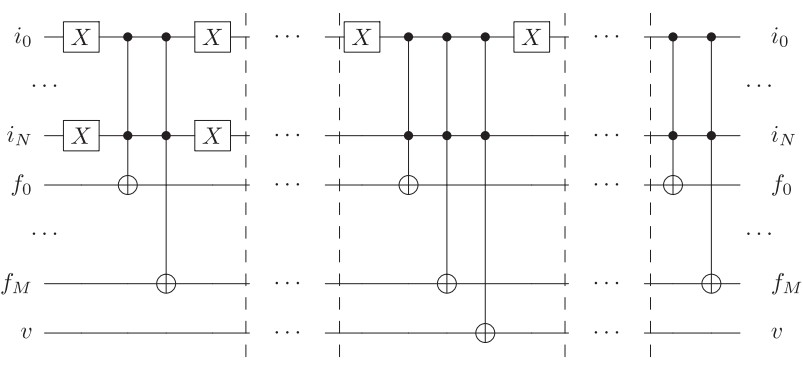

(b) Gate-level implementation of the $U_{\text{fit}}$ subcircuit.

**Figure 3** **The gate-level implementation of the $U_{fit}$ in Grover's algorithm sub-circuit utilizes of $n$-qubits Toffoli gates as presented in** *Nielsen & Chuang (2002)*. The qubits from the individual register are control qubits, while the qubits from the fitness registers are the target qubits. $v$ is the valid qubit used for indicating the validity of the corresponding chromosome.

---

**Algorithm 2**  $U_{fit}$ **sub-circuit description.**

1: Create the individual quantum register $|u\rangle$

2: Create the fitness quantum register $|\,fitness_u\rangle$

3: Create the quantum sub-circuit $U_{f\,it}$

4: **for each** individual in population **do**

5:      fitness_value = calculate_fitness(adjacency_matrix, individual)

6:      **for** $i = 0$ to $M-1$ **do**

7:         **if** fitness_value[i] = 1 **then**

8:            $|\,fitnessu\rangle_i = CNOT(|u\rangle, |\,fitness_u\rangle_i)$

9:         **end if**

10:      **end for**

11:      **if** fitness value is valid **then**

12:         $|\,fitness_u\rangle = CNOT(|u\rangle, |\,fitnessu\rangle valid)$

13:      **end if**

14: **end for**

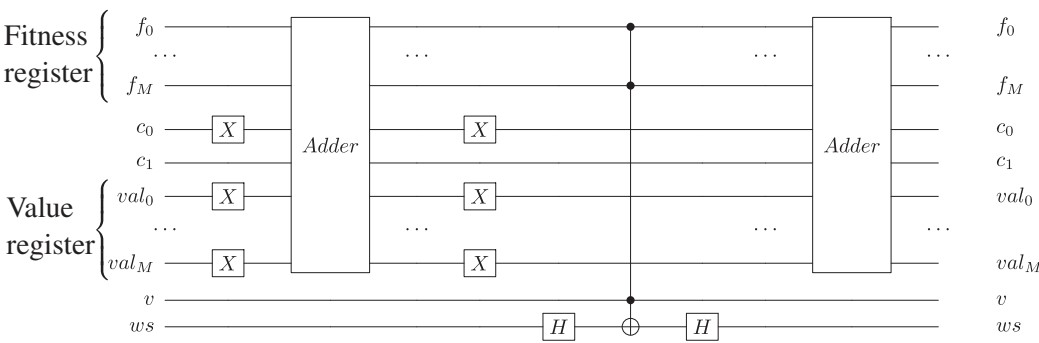

**Figure 4 Oracle circuit made with two quantum two's complement adders, two Hadamard gates and one *n*-qubit Toffoli gate: *f* is the fitness quantum register, *val* is the quantum register storing the *max* value, while $c_0$ and $c_1$ are the carry qubits used in the subtraction and addition circuits; *v* is the valid qubit used to indicate the validity of the corresponding chromosome.** Finally, *ws* is the oracle workspace qubit as presented in *Udrescu, Prodan & Vlăduțiu, 2006*. To perform subtraction, we first prepare the negative value in *val* quantum register, by applying the *X*-gate, and by setting $c_0 = 1$.

The Oracle operates on the fitness register qubits except the validity qubit *v* and uses two's complement number representation for marking the states. As such, by adding −(*max* + 1), (when $max \in \mathbb{N}, max > 0$) to the fitness value, the most significant qubit will always be 0. The corresponding basis states are marked by shifting their amplitudes and the fitness value is restored by adding *max* + 1. This way, the oracle $\tilde{O}_{max}(f_{fit}(u))$ is applied

$$|\psi\rangle_3 = \tilde{O}_{max}|\psi\rangle_2 = (-1)^{g(u)} \frac{1}{\sqrt{2^N}} \sum_{i=0}^{2^N-1} |u\rangle_i \otimes |f_{fit}(u)\rangle_i \qquad (12)$$

such that

$$g(u) = \begin{cases} 1 & if |f_{fit}(u)\rangle_i \geq max \\ 0 & otherwise, \end{cases} \qquad (13)$$

then the corresponding $|f_{fit}(u)\rangle_i$ basis states are marked.

In our algorithm, the oracle is implemented as described in *Udrescu, Prodan & Vlăduțiu (2006)*, using a quantum two's complement subtractor and a quantum two's complement adder, and it is then applied on the entire fitness register (except for the validity qubit). By using the two's complement addition, the correlation between an individual and its corresponding fitness value is not affected because—as presented in *Udrescu, Prodan & Vlăduțiu (2006)*—the addition is a pseudo-classical permutation function. Figure 4 shows the Oracle circuit implementation, while Algorithm 3 provides the pseudocode for the Oracle description. All basis states for which the fitness value is greater than the number of edges are marked by multiplying their amplitudes with −1. For the subtractor and adder implementation, we use a Quantum Ripple Carry Adder circuit, as presented in *Cuccaro et al. (2004)*. The gate-level implementation of the adder subcircuit is presented in Fig. 5. We analyzed the possibility of using a Quantum Carry Look-Ahead Adder (*Cheng & Tseng, 2002*), but it presented the disadvantage of a higher

**Algorithm 3** Oracle subcircuit description.

1: Create $|pqreg\rangle$, a quantum register that stores a positive value representing the maximum number of edges

2: Create $|fitness_u\rangle$–the fitness quantum register

3: Create $|ws\rangle$–the oracle workspace quantum register

4: Create $|cin\rangle$–the carry-in quantum register

5: Create $|cout\rangle$–the carry-out quantum register

6: Create the *Oracle* quantum sub-circuit

7: Append Adder($|pqreg\rangle$,$|fitness_u\rangle$),$|cin\rangle$,$|cout\rangle$ sub-circuit ▷ Subtract the number of edges from the fitness value

8: $|ws\rangle = H|ws\rangle$

9: $|ws\rangle = CNOT(|fitness_u\rangle,|ws\rangle)$

10: $|ws\rangle = H|ws\rangle$

11: Append Adder($|pqreg\rangle$,$|fitness_u\rangle$),$|cin\rangle$,$|cout\rangle$ sub-circuit ▷ Add the number of edges to the fitness value

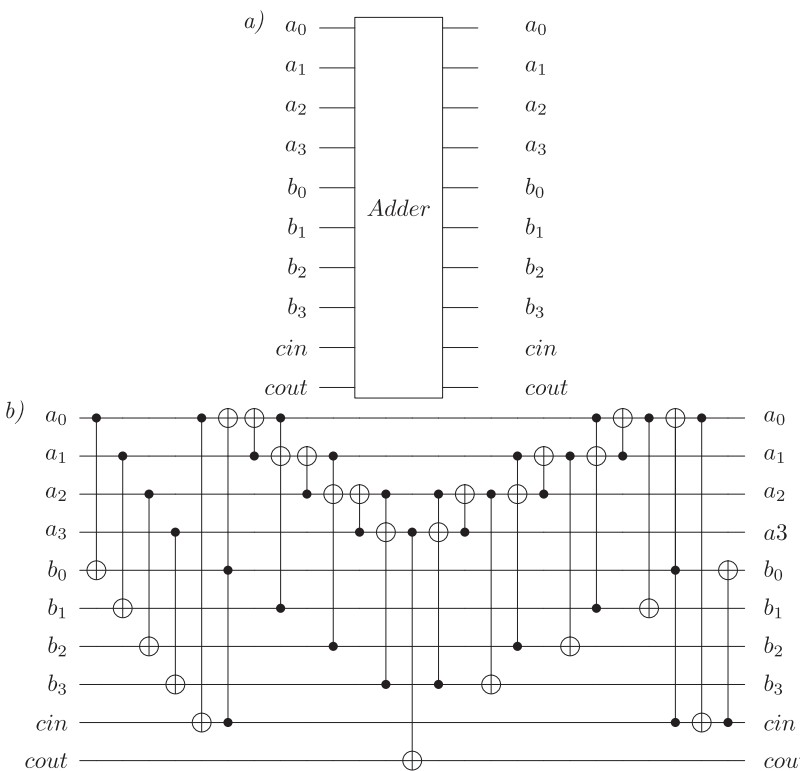

**Figure 5 Four-qubits Quantum Ripple Carry Adder circuit, as presented in *Cuccaro et al. (2004)*.** (A) The adder sub-circuit with the two operands *a* and *b* and the carry qubits. The result of the addition is stored in *b*. Both operands are represented on 4-qubits. (B) The gate-level implementation of the sub-circuit.

**Figure 6 Diffuser sub-circuit.** (A) The sub-circuit with input and output qubits; (B) the gate-level implementation.

number of qubits. Considering $M$ the size of the fitness register and $m = \mathcal{O}\left(\sqrt{2^M}\right)$ the number of Grover Iteration, by using the Ripple Carry Adder circuit we utilize one carry-in qubit and 2 qubits for carry-out in each iteration (1 carry-out qubit for each adder), thus a total of $2\sqrt{2^M} + 1$ qubits. For the Quantum Carry Look-Ahead Adder, we need a total of $2(M + 1)$ carry qubits in each iteration ($M + 1$ carry qubits for each adder), hence not an acceptable solution to be used in our implementation.

Next, we use the Grover diffuser, **G**, to augment the amplitude of the marked states, $|\psi\rangle_i = |f_{fit}(u)\rangle_i$ with $f_{fit}(u) \geq 0$, in the fitness register:

$$|\psi\rangle_4 = G|\psi\rangle_3. \tag{14}$$

We update the value of *max* with the state found. Figure 6 shows the Grover diffuser sub-circuit and its implementation; $f$ is the fitness quantum register, $v$ is the valid qubit and $ws$ is the oracle workspace. In our implementation, we used the Hadamard Gate, the Pauli-X gates and the $n$-qubit Toffoli gate.

The last step in the algorithm is to measure $|\psi\rangle_{m-1}$ register to obtain the corresponding individual. The measured value represents the solution found to solve the problem.

## Complexity analysis

RQGA, as presented in *Udrescu, Prodan & Vlăduțiu (2006)*, is an adaptation of the quantum maximum finding algorithm introduced in *Ahuja & Kapoor (1999)*. According to *Ahuja & Kapoor (1999)*, the algorithm requires $\mathcal{O}\left(\sqrt{N}\right)$ queries made to the oracle. Since in RQGA the initial state processed using Grover's Algorithm is an equally weighted superposition, it means that no extra Grover Iterations are required in order to augment the amplitudes (*Udrescu, Prodan & Vlăduțiu, 2006*)—the algorithm maintains the number of steps of the quantum maximum finding algorithm. Thus, RQGA's complexity is $\mathcal{O}\left(\sqrt{N}\right)$, where $N$ is the search space size.

For solving the graph coloring problem using RQGA, we define a $M$-qubit fitness quantum register. As shown, the size of the search space is $2^M$. Therefore, according to *Grover (1996)* and *Nielsen & Chuang (2002)*, the number of operations in which the oracle is consulted is $\mathcal{O}\left(\sqrt{2^M}\right)$. Considering that there are $N$ possible solutions to the problem, according to *Nielsen & Chuang (2002)*, the algorithm only needs to consult the oracle $\mathcal{O}\left(\sqrt{\frac{2^M}{N}}\right)$; thus, the complexity of the algorithm in solving the graph coloring problem is

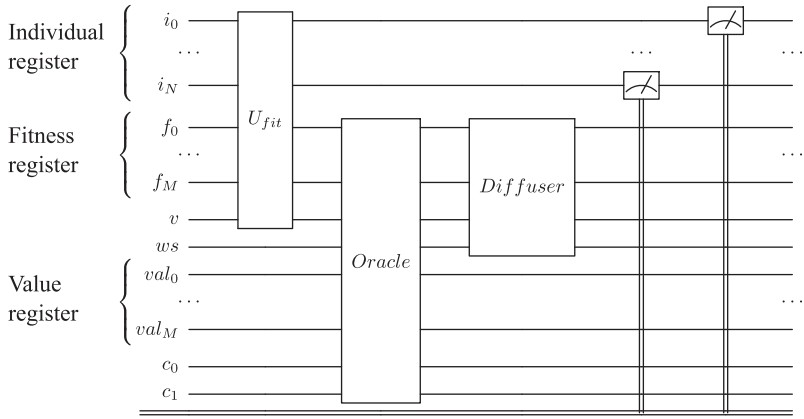

**Figure 7 Reduced quantum genetic algorithm implementation for solving the graph coloring problem.** The individual quantum register is represented by *i*, the fitness quantum register by *f*, while *v* is the valid qubit; *val* represents the *max* value as presented in *Udrescu, Prodan & Vlăduțiu, 2006*. The carry-in and carry-out qubits used by adder sub-circuits are represented by $c_0$ and $c_1$. We represent only one Grover Iteration for simplicity.

$\mathcal{O}\left(\sqrt{\frac{2^M}{N}}\right)$ if there are *N* possible solutions.

## MEASUREMENTS AND RESULTS

For performing the measurements, we implemented the algorithm as presented in Fig. 7, using Qiskit—for the simulation we use the **IBMQ** back end (*Anis et al., 2021*). We performed the simulations using the *ibm-q* provider with the *ibmq_qasm_simulator* back end (version 0.1.547 with a configuration of 16 shots). The simulator is general and context-aware—a general-purpose simulator for simulating quantum circuits, both ideally and subject to noise modeling—limited to 32-qubit circuits. The following basic gates are available on the mentioned simulator: U1, U2, U3, U, P, R, RX, RY, RZ, ID, X, Y, Z, H, S, SDG, SX, T, TDG, SWAP, CX, CY, CZ, CSX, CP, CU1, CU2, CU3, RXX, RYY, RZZ, RZX, CCX, CSWAP, MCX, MCY, MCZ, MCSX, MCP, MCU1, MCU2, MCU3, MCRX, MCRY, MCRZ, MCR, MCSWAP, UNITARY, DIAGONAL, MULTIPLEXER, INITIALIZE, KRAUS, ROERROR, DELAY. In the implementation of RQGA we used Hadarmard, Pauli-X, Toffoli, and *n*-qubits Toffoli gates which correspond to the H, X, CCX and MCX gates available in the simulator.

To study the complexity of the algorithm, we run it on 12 data sets. We utilize 10 data sets for testing the node coloring; half of them are represented by 4-node randomly generated graphs while the rest are 5-node randomly generated graphs. The remaining two data sets are used for testing the edge coloring capabilities of the algorithm. Each graph is colored using a maximum of 3 colors, *K* = 3. For the fitness representation, we use 4 qubits, thus, the number of Grover Iterations needed for finding the results is $\mathcal{O}\left(\sqrt{2^4}\right) \approx 4$. To analyze the complexity of the algorithm, we variate the number of Grover Iterations for each data set and perform each measurement 10 times. We are interested in observing the convergence of the algorithm, and for this purpose we analyze

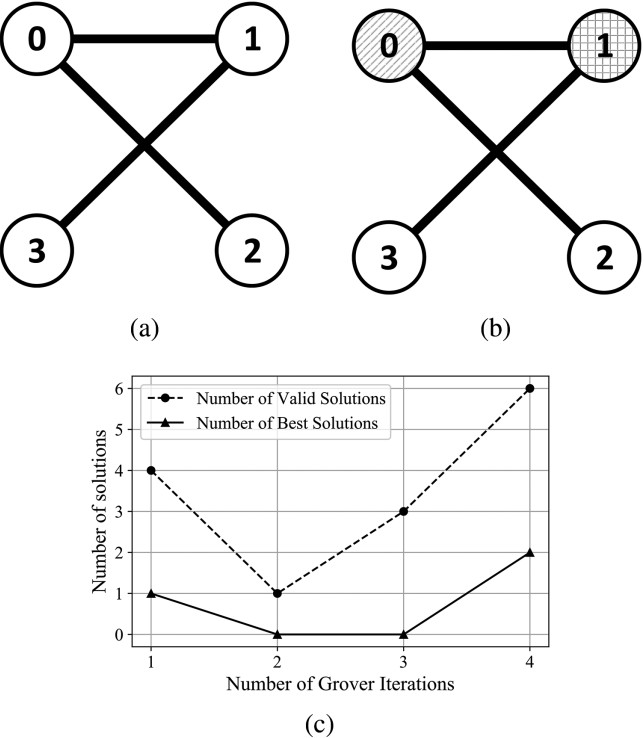

**Figure 8** (A) An Erdős-Rényi graph generated with edge probability 0.4 and 4 nodes, which we used for the coloring problem. (B) The same graph colored with the solution found by our algorithm. The result for this graph uses three colors, so that nodes 3 and 2 have the same color and nodes 0 and 1 are colored using different colors. (C) The experimental results; after four iterations the algorithm produced six valid solutions from which two are best solutions.

the solutions that represent the local optimum (*i.e.*, valid solutions) and those that represent the global optimum (*i.e.*, best solutions) for the coloring problem.

We start by presenting the results obtained after running the node coloring algorithm on 4-node graphs and continue with the results obtained after running our algorithm on 5-node graphs. The presented outcomes are ordered by the number of edges. Last, we show the results obtained after running the edge coloring algorithm.

## Experimental results

In Supplemental Information, Figs. S1, S2, and S3 we notice that the algorithm finds only the best solutions, starting from the first Grover Iteration. As expected, we achieve the best outcome in terms of the number of best solutions after 4 Grover iterations. In these cases, our algorithm also found the chromatic number.

For the graphs in Fig. S4 from Supplemental Information, and Fig. 8—graphs with a higher number of edges–the algorithm found valid and best solutions, with the best outcome (in terms of number of solutions) obtained after 3 and 4 Grover iterations.

For the graphs presented in Supplemental Information, Figs. S5 and S6, the algorithm determines both the best coloring and chromatic number, with best outcome (in terms of number of solutions) obtained after 3 and 4 Grover iterations. In the case of graphs

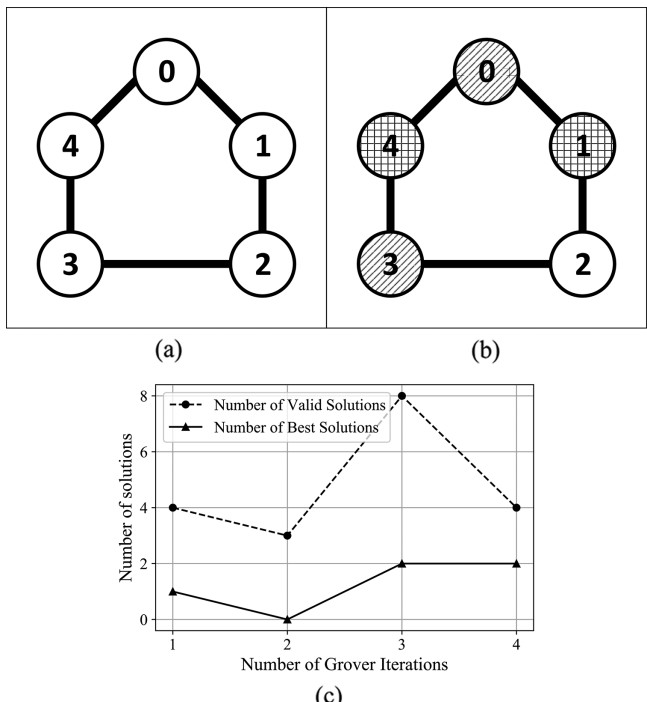

**Figure 9** (A) An Erdős-Rényi graph generated with edge probability 0.4 and five nodes, which we used for the coloring problem. (B) The same graph colored with the solution found by our algorithm. The solution for this graph uses three colors, so that nodes 0 and 3 are colored using the first color, nodes 1 and 4 are colored using the second color, node 2 is colored using the third color. (C) The experimental results; after 3 iterations the algorithm produced eight valid solutions from which two are best solutions.

presented in Figs. S7 and S8 from Supplemental Information, and Fig. 9—graphs with a higher number of edges—the algorithm determined the best coloring after 2 Grover iterations.

In Figs. 10 and 11, we present the relationship between the number of Grover iterations and the minimum, maximum, and average number of valid and best solutions found after testing the node coloring algorithm on graphs with 4 and 5 nodes.

In Figs. 12 and 13, we present the results after using the algorithm to solve the edge coloring problem. For the graph presented in Fig. 12, the algorithm determined the best results after 1 Grover iteration. In Fig. 3, the algorithm finds the best solution after 2 Grover iterations. We obtain the best outcome from the point of view of valid and best solutions, after 3 oracle queries.

## DISCUSSIONS

We observed experimentally that, for the 4-node graphs, some of the best solutions found used only 2 colors instead of the configured 3 colors. Thus, without any modifications in the fitness function for simple graphs, the algorithm is capable of determining both the coloring and the chromatic number. For more complex graphs, the algorithm can determine the chromatic number by modifying the fitness function: for the valid individuals that color the graph using a number of colors smaller than the configured one,

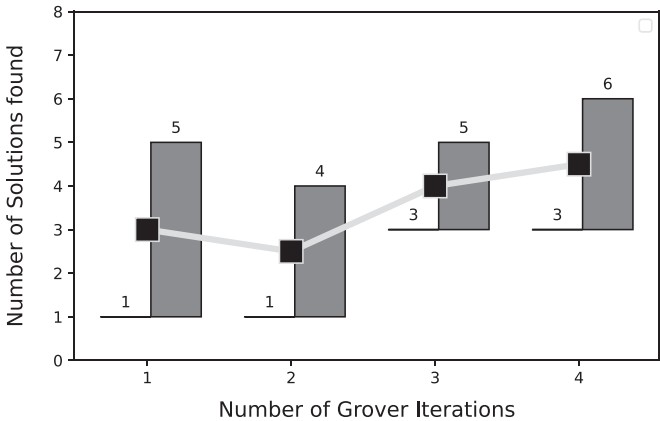

(a) The relationship between the number of Grover iterations and the minimum, maximum, and average number of valid solutions.

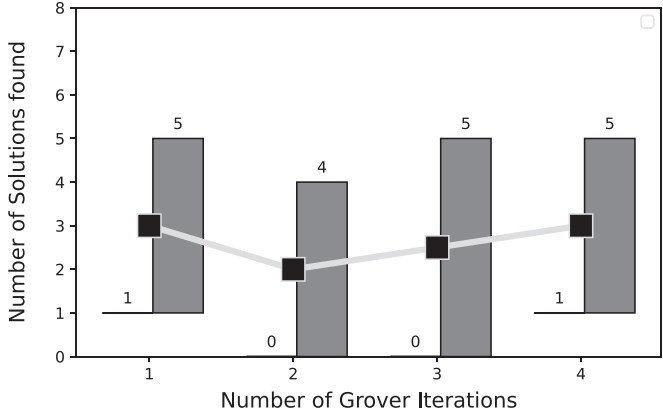

(b) The relationship between the number of Grover iterations and the minimum, maximum, and average number of best solutions.

**Figure 10 The solutions for the graph coloring problem described in subsection problem statement.** For each number of Grover iterations, both panels depict the minimum, maximum, and average number of solutions extracted from the results obtained after coloring the 4-node graphs using our algorithm.

the fitness value will be amplified by a factor. Thus, solutions with a better (*i.e.*, smaller) chromatic number will have a bigger fitness value, which generates a higher probability to be chosen as the best solutions.

In subsection *Complexity analysis*, we show that the expected complexity is $\mathcal{O}\left(\sqrt{\frac{2^M}{N}}\right)$ oracle queries, where $M$ is the fitness quantum register size and $N$ is the number of possible solutions. Experimentally, the complexity assessment is confirmed; in most of the experiments, the algorithm found best solutions to the problem after the first queries, confirming that only $\mathcal{O}\left(\sqrt{\frac{2^M}{N}}\right)$ iterations are required.

Compared to other solutions, our approach using RQGA can be used to solve both the node and edge coloring forms of the graph coloring problem. Moreover, with minimal adjustments to the fitness function, our solution can determine the chromatic number. In

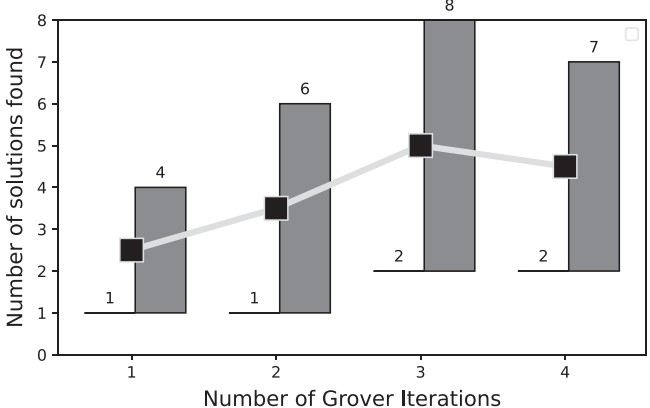

(a) The relationship between the number of Grover iterations and the minimum, maximum, and average number of valid solutions.

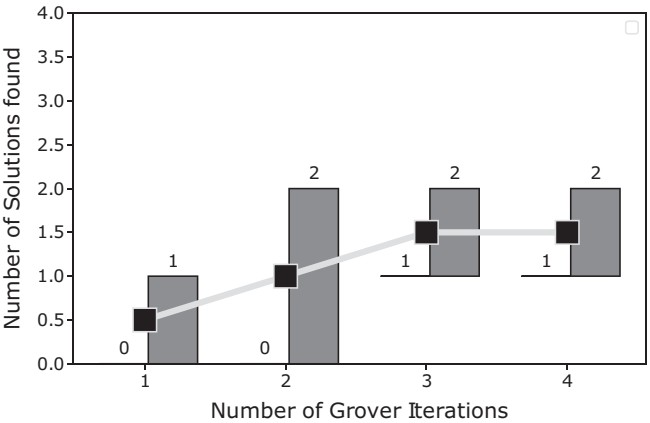

(b) The relationship between the number of Grover iterations and the minimum, maximum, and average number of best solutions.

**Figure 11** **The solutions for the graph coloring problem described in subsection problem statement.** For each Grover iteration, both panels depict the minimum, maximum, and average number of solutions extracted from the results obtained after coloring the 5-node graphs with our algorithm.

Table 2, we provide a comparison between our approach and other algorithms. Since the literature presents different types of heuristic algorithms, a comparison from the point of view of algorithm convergence is not relevant. Nevertheless, a possible comparison approach is to use the perspective of the number of solutions and optimality. Such comparisons with the heuristic algorithms presented would involve running RQGA on the same data sets. Unfortunately, since our algorithm is simulated on a classical computer, we have limitations on the number of qubits we can simulate, owing to the simulation time and circuit complexity (*i.e.*, number of Controlled-Not gates). Since our approach is purely quantum, we need to compare it with purely quantum algorithms. Thus, we first consider a comparison that focuses on the forms of graph coloring problem that are solved, the ability to determine the chromatic number, and metrics used in order to determine the algorithm's performance. Additionally, we provide a comparison that focuses on the solution count between RQGA and the algorithms presented in *Silva et al.*

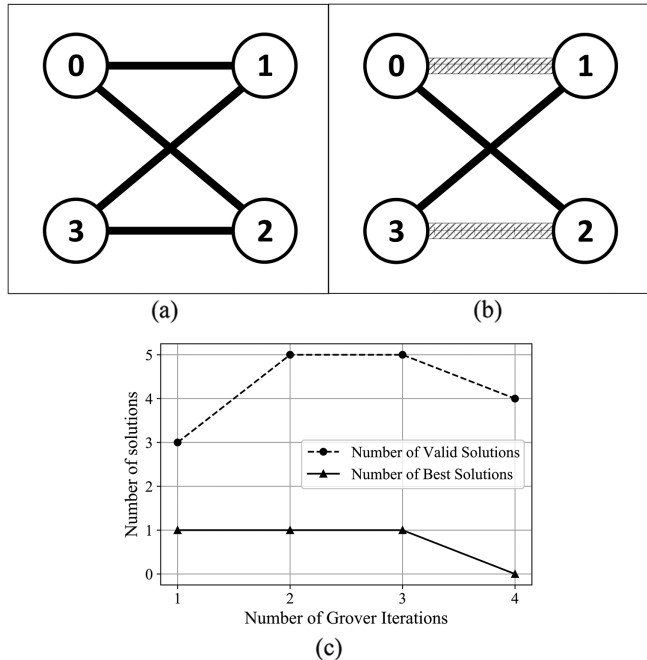

**Figure 12** (A) An Erdős-Rényi graph generated with edge probability 0.4 and four nodes, which we used for the coloring problem. (B) The same graph with edge colored using the solution found by our algorithm. The solution for this graph uses two colors. (C) The experimental results: after three iterations, the algorithm produced five valid solutions from which 1 is best solution.

*(2020)*; we also compare our approach with the algorithms presented in *Silva et al. (2020)*, *Tabi et al. (2020)*, and *Aragón Artacho & Campoy (2018)* from the perspective of the number of iterations required to find a solution that colors a 5-node graph.

We performed 3 simulations with 8,192 repetitions to count the number of solutions found by our approach that color the graph from Fig. 9, as presented in Fig. 14. In Fig. 15 we present the comparison between our approach and the algorithms presented in *Silva et al. (2020)* that focus on the ratio between the number of solutions (optimal, possible, and none) and the number of repetitions. RQGA performed 4 iterations 8,192 times while the algorithms presented in *Silva et al. (2020)* performed 5,000 iterations 10,000 times. For all algorithms, we divided the number of solutions found by the number of iterations multiplied with the number of repetitions. Therefore, as presented in Fig. 15, RQGA performed better than the other algorithms because our approach has a success probability of 0.76% for finding the optimal solutions, compared to 0.0027% representing the best of the algorithm $C\_Q\_sim$.

In Fig. 16 we present a comparison between RQGA and the algorithms presented in *Silva et al. (2020)*, *Tabi et al. (2020)*, and *Aragón Artacho & Campoy (2018)*, from the perspective of the number of iterations required to find the solutions that color a 5-node graph. As presented in subsection *Complexity analysis* and confirmed experimentally, RQGA requires 4 iterations for finding the solutions while the mentioned heuristic algorithms require 44, 5,000, and 100 iterations, respectively.

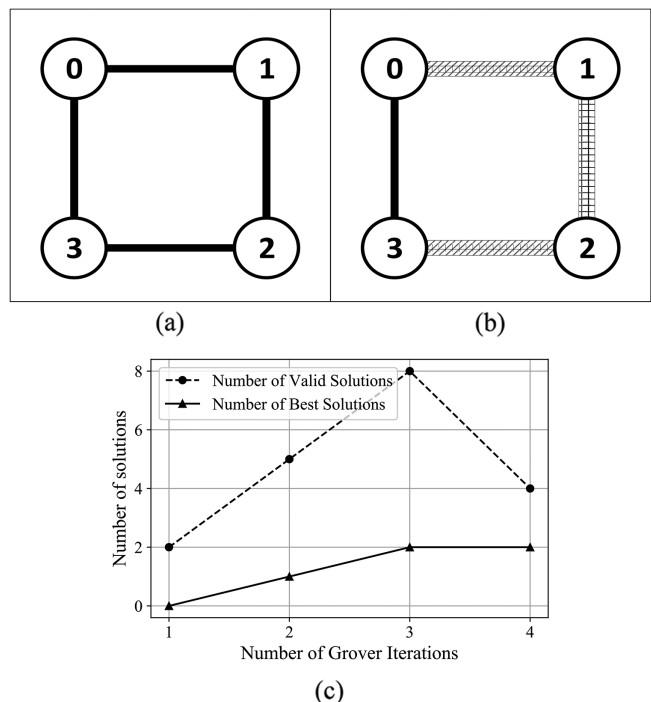

**Figure 13** (A) An Erdős-Rényi graph generated with edge probability 0.4 and four nodes, which we used for the coloring problem. (B) The same graph with edge colored using the solution found by our algorithm. The solution for this graph uses three colors. (C) The experimental results: after three iterations, the algorithm produced eight valid solutions from which 2 are best solution.

**Table 2** Comparison between different approaches in solving the graph coloring problem. Since we include different heuristic algorithms, the comparison focuses on characteristics such as: graph coloring form, ability to determine the chromatic number, metrics used in order to determine algorithm's performance.

| Algorithm | Graph coloring form | Chromatic number | Observed metrics |
|---|---|---|---|
| RQGA | Node and edge coloring | Minimal adjustments | $\mathcal{O}\left(\sqrt{2^M}N\right)$ complexity |
| Quantum optimization for the graph coloring problem with space-efficient embedding *Tabi et al. (2020)* | Node coloring | Not mentioned | Convergence and CPU time |
| Graph coloring with quantum annealing *Kwok & Pudenz (2020)* | Node coloring | Not mentioned | Success probability, time to solution, wallclock time |
| Graph coloring with quantum heuristics *Fabrikant & Hogg (2002)* | Node coloring | Not mentioned | Asymptotic analysis |
| Quantum annealing of the graph coloring problem *Titiloye & Crispin (2011)* | Node coloring | Not mentioned | Number of solutions found |
| Constrained quantum annealing of graph coloring *Kudo (2018)* | Node coloring | Not mentioned | Residual energy and success probability |
| Mapping graph coloring to quantum annealing *Silva et al. (2020)* | Node coloring | Not mentioned | Number of optimal solutions |

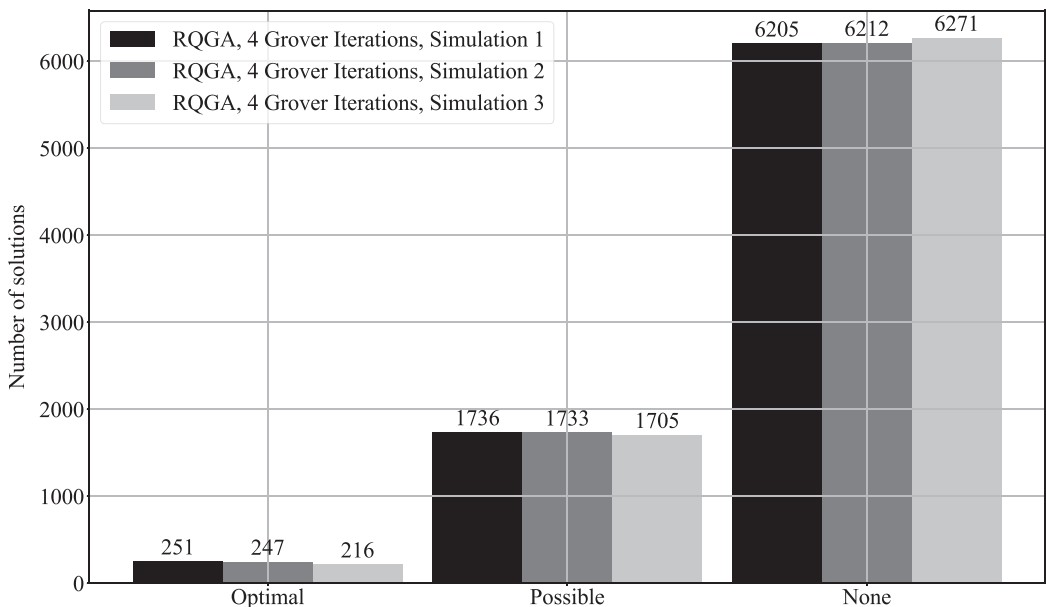

**Figure 14 Solutions count for RQGA. We performed three simulations with 8,192 repetitions to solve the coloring of the graph from Fig. 9.** We use the notations from *Silva et al. (2020)* to denote best, valid, and invalid solutions. As such, optimal represents best solutions, valid solutions are called possible, and the invalid ones are called none.

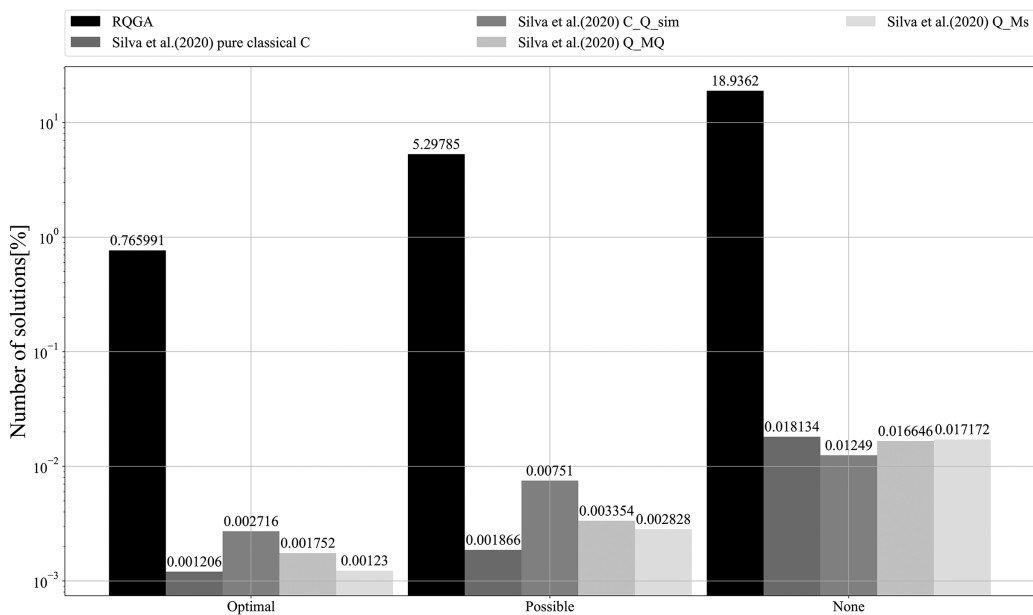

**Figure 15 Comparison between RQGA and the algorithms presented in *Silva et al. (2020)*.** All algorithms solved the coloring of the graph from Fig. 9. We use the notations from *Silva et al. (2020)* to denote best, valid, and invalid solutions. As such, optimal represents best solutions, valid solutions are called possible, and the invalid ones are called none.

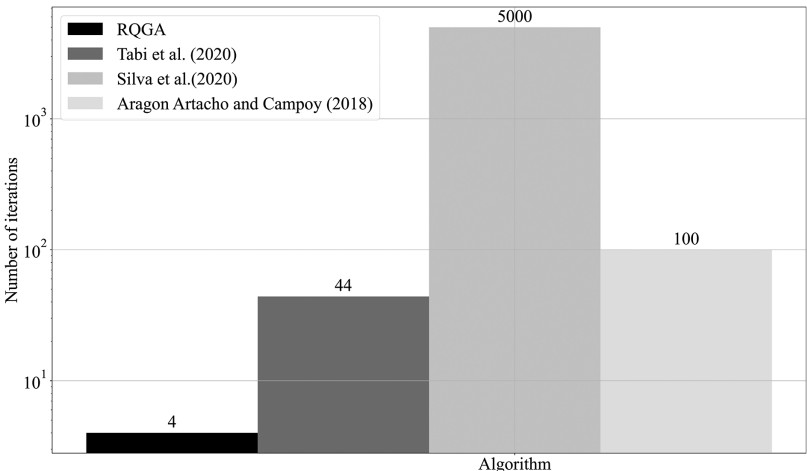

**Figure 16 Comparison between RQGA and the algorithms presented in** *Silva et al. (2020)*, *Tabi et al. (2020)*, **and in** *Aragón Artacho & Campoy (2018)* **from perspective of number of iterations required for finding the solutions that colors an 5-nodes graph.** Compared to the mentioned algorithms, our approach needs only four iterations for coloring the graph.

## LIMITATIONS

Our study is limited by the fact that we propose a purely quantum method that needs to be simulated, given that we do not have access to a quantum computer. As mentioned in *Udrescu, Prodan & Vlăduțiu (2012)*, the simulation of a quantum circuit on classical computers requires run-times exponential with the circuit size. *Spector (2004)*, shows that the space and time requirements for simulation are of the order $2^n$ for a system with $n$-qubits. Moreover, *Viamontes, Markov & Hayes (2009)* argues that the linear algebraic simulation of quantum computers would have time and memory complexity $\mathcal{O}(2^{2n})$ for an $n$-qubit system. Indeed, due to the fourth postulate of quantum mechanics, the simulation of quantum behavior on classical computers has exponential complexity.

Consequently, since we simulated the algorithm, we have limitations on the number of qubits, simulation time, and circuit complexity. Such limitations have an impact on the size of the datasets on which we apply the algorithm. Knowing the number of nodes in a graph, the number of colors available, and the number of Grover iterations, we calculate with function $f$ the number of qubits required by the algorithm

$$f(|V|, n, M, m) = |V| \times n + 2 \times (M + m) + 3. \tag{15}$$

In Eq. (15), variable $|V|$ represents the number of nodes in graph, $n$ represents the number of qubits needed for color representation (as presented in subsection *Implementation*); $M$ represents the number of qubits required by fitness representation in two's complement; $m = \mathcal{O}\left(\sqrt{2^M}\right)$ represents the number of Grover Iterations; 3 qubits are needed for carry-in, oracle workspace, valid flag, while adders need 2 qubits for carry-out in each Grover iteration.

As such, for the graph presented in the Supplemental Information, Graph coloring example—with 4 nodes—2 qubits are used for color representation, and 4 qubits are

used for fitness value representation. Considering that we need 4 Grover iterations, the circuit requires a total of $f(4,2,4,4) = 4 \times 2 + 2 \times 8 + 3 = 27$ qubits. Therefore, according to *Spector (2004)*, the space and time requirement for the simulation of the circuit is $2^{27}$.

An increased number of qubits have an exponential impact on the circuit complexity and on the simulation time of the algorithm. Thus, considering the above-mentioned limitations, we could not simulate the algorithm on more complex graphs (*i.e.*, with a high number of nodes and edges).

## CONCLUSIONS

This paper describes the instantiation of the Reduced Quantum Genetic Algorithm for solving the graph coloring problem with the Qiskit toolchain. By this means, we introduce a pure quantum heuristic method that solves the problem in $\mathcal{O}\left(\sqrt{\frac{2^M}{N}}\right)$ Grover oracle queries. We also provide solutions for the graph node coloring problem, the edge coloring problem, and finding the chromatic number of a graph with several fitness adjustments.

Because of the big number of qubits required by the algorithm implementation, our experiments based on computer simulations are limited. We observed the results to determine the convergence and complexity of the algorithm by analyzing the solutions for the problem. Experimentally, we prove the complexity assessment and we demonstrate that, at least for graphs with a small number of nodes, the algorithm not only finds the best solutions for the problem, but also the chromatic number.

Future research can analyze how the complexity of the algorithm can be reduced by employing meta-heuristics, as well as the impact of meta-heuristics on convergence. Indeed, our focus will be on finding a way to reduce the number of oracle queries, such that the complexity of the algorithm is reduced without affecting convergence.

## ACKNOWLEDGEMENTS

We acknowledge the use of the IBM Q for this work. The views expressed are those of the authors and do not reflect the official policy or position of IBM or the IBM Q team.

## ACRONYMS

| | |
|---|---|
| **COA** | Cuckoo Optimization Algorithm. 5 |
| **CQA** | Constrained Quantum Annealing. 5 |
| **GA** | Genetic Algorithm. 1, 4 |
| **HGA** | Hybrid Genetic Algorithms. 4 |
| **QA** | Quantum Annealing. 5, 6 |
| **QAOA** | Quantum Approximate Optimization Algorithm. 5 |
| **QGA** | Quantum Genetic Algorithm. 4 |
| **QGOA** | Quantum Genetic Optimization Algorithm. 4 |
| **QIEA** | Quantum-Inspired Evolutionary Algorithm. 4 |
| **QIEVSA** | Quantum-Inspired Evolutionary View Selection Algorithm. 4 |
| **QIGA** | Quantum-Inspired Genetic Algorithm. 4 |
| **QRAM** | Quantum Random Access Memory. 6 |

| QUBO | Quadratic Unconstrained Binary Optimization Problem. 6 |
|------|------|
| **RQGA** | Reduced Quantum Genetic Algorithm. 1–4, 6, 7, 14, 15, 22–25 |
| **TLBO** | Teaching-Learning-Based Optimization. 5 |

### Funding
The authors received no funding for this work.

### Competing Interests
The authors declare that they have no competing interests.

### Author Contributions
- Sebastian Mihai Ardelean conceived and designed the experiments, performed the experiments, analyzed the data, performed the computation work, prepared figures and/or tables, authored or reviewed drafts of the paper, and approved the final draft.
- Mihai Udrescu conceived and designed the experiments, analyzed the data, prepared figures and/or tables, authored or reviewed drafts of the paper, and approved the final draft.

### Data Availability
The data is available at GitHub: https://github.com/sebastianardelean/graphcoloringusingrqga.

### Supplemental Information
Supplemental information for this article can be found online at http://dx.doi.org/10.7717/peerj-cs.836#supplemental-information.

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
