# Peer review of "Graph coloring using the reduced quantum genetic algorithm"

_PeerJ Computer Science, doi:10.7717/peerj-cs.836_

## Round 0.1 · original submission · Major Revisions

You are advised to incorporate the comments of the reviewers.

·

Basic reporting

The manuscript aims to solve the graph coloring problem using the Reduced Quantum Genetic Algorithm. The general structure of the manuscript is respectable with good professional English and clear links between sections. However, there are some weaknesses in the basic reporting that must be revised, from which we can briefly cite the following points:
The list of references, especially in the literature review section needs to be updated. Authors are advised to use more recent references.
Some terms and definitions need to be detailed. For instance, quantum computing is not well defined and detailed although it is a fundamental concept in the solving approach.
Some assumptions were cited by the authors without any explanation or justification. For example, authors said in lines 37 and 38 ‘On the other hand, quantum computation is very powerful for solving various problems due to its 38 specific properties and phenomena such as entanglement, interference, and exponential parallelism. (Need to add a reference to justify).

Experimental design

The results section is very short and does not contain depth analysis to prove the efficiency of the proposed approach. No Rigorous investigation was performed in the manuscript. The results section needs to be revised with consideration to the following points:
• To add a comparative study of the results (and not metrics) with similar solving approaches applied to the same problem (the graph coloring problem).
• To perform a statistical analysis to validate the high performance of the proposed approach.
• Apply the solving approach to more practical, complex, real-life problems and not only the graph coloring problem.

Validity of the findings

No innovative points and contributions can be easily found in the proposed approaches. Authors clearly cited in the Methodology section (lines 162, and 163, 167 and 168) that they will use the Reduced Quantum Genetic Algorithm proposed by Udrescu et al. (2006), So what is the contribution of the authors? Authors do not present their contribution clearly - where is the novelty comparing to existing methods? In addition, the proposed solving algorithm must beapplied to more applicable, real-life problems to prove its effeciency and strengh.

·

Basic reporting

Need Clear Definitions

Experimental design

The authors have to compare the proposed algorithm with more number of existing algorithms

Validity of the findings

Novelty not assesed

Additional comments

Review Comments:
In this work, the authors proposed the reduced quantum genetic algorithm for graph coloring. Although, the approach could be of interest but there are some major concerns which should be addressed.
1. The Figure 8 and 9 are not clear. I suggest the authors to provide the axis titles that makes more understanding of the figures for readers.
2. In Line No.281, the authors mentioned “adders need 2 qubits” and in Line No: 283, the authors mentioned “4 qubits are used for fitness value representation”. How the parameter values are set to specific threshold value? The authors have to justify the parameter values.
3. In Line No. 311, the authors mentioned that the proposed algorithm determines the chromatic number in Figures 17 and 18. But the details are not explained in the Figures description.
4. The authors didn’t mention about the system specification.
5. The authors should make available their software with documentation for its usage and examples for testing in a public repository like GitHub.
6. Limitations of the proposed study have to be included in the manuscript.
7. The authors shown the experimental results for their proposed algorithm. However, they have not compared with other recent optimization algorithms such as Modified Cuckoo algorithm, ABC optimization algorithm, Memetic Teaching–Learning-Based Optimization algorithm and so on. I suggest the authors to compare the proposed algorithm with more number of existing algorithms.
• Mahmoudi, Shadi & Lotfi, Shahriar. (2015). Modified Cuckoo Optimization Algorithm (MCOA) to solve Graph Coloring problem. Applied Soft Computing. 33. 10.1016/j.asoc.2015.04.020.
• Dökeroğlu, Tansel & Sevinç, Ender. (2021). Memetic Teaching-Learning-Based Optimization Algorithms for Large Graph Coloring Problems. Engineering Applications of Artificial Intelligence. 102. 10.1016/j.engappai.2021.104282.
• R. S. Tomar, S. Singh, S. Verma and G. S. Tomar, "A Novel ABC Optimization Algorithm for Graph Coloring Problem," 2013 5th International Conference and Computational Intelligence and Communication Networks, 2013, pp. 257-261, doi: 10.1109/CICN.2013.61.
• Aragón Artacho, F.J., Campoy, R. Solving Graph Coloring Problems with the Douglas-Rachford Algorithm. Set-Valued Var. Anal 26, 277–304 (2018). https://doi.org/10.1007/s11228-017-0461-4

8. I suggest the authors to provide the significance of the proposed algorithm to the society.

This manuscript standard is not suitable for publication in this esteemed journal. Hence, the manuscript needs “major revision” for further steps.

---

## Round 0.2 · accepted · Accept

Congratulations for your efforts.

·

Basic reporting

The authors have addressed the concerns I have raised adequately (expect the third comment of the Experimental Design section) and they have improved the paper largely. I think the paper is now accepted and publishable.

Experimental design

The experimental design was improved.

Validity of the findings

The findings section was improved by using the comparative study.